# Self-Supervised Neural Regression for High-Precision Geometric Alignment

## Abstract

We propose a Self-Supervised Machine Learning framework for the high-precision geometric alignment of large-scale physical systems with mechanical misalignment. The system comprises a large-scale high-granularity array of independent sensors, each defined by a six-degree-of-freedom (6-DoF) geometric alignment model. To determine the optimal correction parameters for each sensor, we simultaneously train a substantial number of differentiable lightweight affine transformation modules. Our core contribution is the formulation of a differentiable Physics-Informed $\chi^2$ cost function, which is derived from the system's physical consistency constraints and real data. This framework operates without data labeling, using the physics principles themselves as a supervisory signal. We applied this approach to the Inner Tracking System of the ALICE experiment at CERN, training the models on a large sample of particle trajectories. Compared with the standard method, our proposed approach leads to a reduced systematic bias and an improved resolution in the Distance of Closest Approach (DCA), which is the primary metric determining the track pointing accuracy to the primary vertex, specifically in critical kinematic regions.

## 1. Introduction

Many scientific and engineering domains, including robotics, autonomous driving, astronomy, and particle physics, rely on large-scale systems composed of tens of thousands of individual sensors. The overall performance of these systems is critically dependent on the precise geometric correction of all components, which requires micron-level accuracy. This correction task presents a large-scale optimization problem.

[1]Anonymous Institution, Anonymous City, Anonymous Region, Anonymous Country. Correspondence to: Anonymous Author <anon.email@domain.com>.

Preliminary work. Under review by the International Conference on Machine Learning (ICML). Do not distribute.

The system under study, the ALICE ITS2 detector (ALICE Collaboration, 2014), consists of $N = 24,120$ individual CMOS Monolithic Active Pixel Sensors (MAPS). Each sensor has six degrees of freedom (6-DoF) to describe misalignment position and orientation (3 translations, 3 rotations), resulting in approximately 150,000 parameters that must be optimized simultaneously.

In high-energy physics, the transition toward high-granularity pixel detectors has been crucial for achieving superior spatial precision. The conventional alignment standard, Millepede (Blobel, 2006), solves linearized least-squares problems for detector deformation. However, as the number of independent sensors grows to achieve such high granularity, the parameter space expands drastically. This exposes a fundamental limitation of the analytical approach and the computational cost of matrix calculation scales super-linearly with the number of parameters. Consequently, performing a full sensor-level alignment becomes computationally expensive in terms of time and memory resources, while remaining heavily dependent on an expert-driven iterative process requiring a priori geometry parametrization.

Motivated by this necessity, we present the core ideas for implementing a lightweight, data-driven framework to effectively perform sensor-level alignment as follows.

1. Utilize the system's underlying physical constraints as a self-supervised signal, eliminating the need for labeled data.

2. Formulate these constraints as a differentiable physics-informed $\chi^2$ cost function that quantifies a geometric inconsistency, enabling end-to-end optimization grounded in physical principles.

3. Optimize a substantial number of sensor-wise regression models estimating the correction parameters through gradient-based training.

4. Parallelize the processing of trajectories, substantially reducing total CPU time and enabling scalable end-to-end training on massive datasets.

We validate this framework on a challenging, large-scale system study: the alignment of the high-precision tracking detector of the ALICE experiment at CERN.

Our contributions are:

1. A self-supervised learning framework for the geometric alignment of large-scale systems, requiring no labeled data while demonstrating intrinsic robustness to significant physics-induced noise.

2. A differentiable physics-informed $\chi^2$ cost formulation that enforces geometric correctness and enables end-to-end optimization grounded in physical principles.

3. A scalable architecture capable of optimizing tens of thousands of sensor-wise regression models in parallel, demonstrating efficient gradient-based learning at realistic operational scales.

4. Quantitative improvements in geometric accuracy, including the removal of systematic biases and the enhanced resolution in key physics-performance metrics.

## 2. Related Work

This new approach is positioned at the intersection of conventional optimization methods, physics-informed machine learning, and self-supervised learning.

### 2.1. Conventional Method

Large-scale alignment problems in the high-energy physics experiments have been addressed primarily through the Millepede algorithm. This is a linearized least-squares minimization method that efficiently solves large matrix equations by separating parameters into global (physical geometry) and local (trajectory data) sets, as detailed in Appendix A. Whereas the traditional method depends on the expert-guided linearization and iteration, our new approach replaces these workflows with a data-driven, regression-oriented, and gradient-based optimization, allowing the entire alignment procedure to be automatically learned from data.

### 2.2. Physics-Informed Machine Learning

Our approach follows the core principle of Physics-Informed Neural Networks (PINNs) (Karniadakis et al., 2021): physical processes are embedded directly into the cost function. In our case, the dominant sources of the trajectory uncertainty, pixel-level measurement resolution and multiple scattering (Workman et al., 2022), are incorporated through a physics-informed $\chi^2$ objective that guides the model toward physically consistent geometry corrections. This integration extends the PINN framework to a large-scale regression problem for the detector alignment.

### 2.3. Self-Supervised Learning

The Self-Supervised nature of our framework means that no labeled data are required. Instead of relying on externally provided supervisory targets, our method derives supervision directly from physics-based consistency requirements. The $\chi^2_{\text{track}}$ term (trajectory smoothness) and the $\chi^2_{\text{vertex}}$ term (DCA-based origin consistency) jointly serve as a physics-driven self-supervisory signal that compels the model to converge toward the correct geometric configuration where all physical constraints are simultaneously satisfied.

## 3. Problem Formulation

This study addresses the geometric alignment of a system composed of $N = 24,120$ individual sensors. Each sensor records position measurements in its local 2D coordinate system $\mathbf{r}_{\text{Local}} \in \mathbb{R}^3$ where the component normal to the sensor plane is defined as zero. These must be mapped to a global 3D coordinate system $\mathbf{r}_{\text{Global}} \in \mathbb{R}^3$ using sensor-specific transformation matrices $(\mathbf{R}, \mathbf{T})$, where $\mathbf{R} \in \mathbb{R}^{3\times3}$ represents the rotation and $\mathbf{T} \in \mathbb{R}^3$ the translation, that describe the detector geometry.

$$\mathbf{r}_{\text{Global}} = \mathbf{R}\mathbf{r}_{\text{Local}} + \mathbf{T}. \tag{1}$$

This representation is effective for identifying and correcting discrepancies between the software-defined sensor positions and the actual hardware installation positions.

$$\mathbf{r}'_{\text{Global}} = \mathbf{R}(\mathbf{r}_{\text{Local}} + \boldsymbol{\Delta}\mathbf{r}_{\text{corr}}) + \mathbf{T}, \tag{2}$$
$$= \mathbf{R}'\mathbf{r}_{\text{Local}} + \mathbf{T}'. \tag{3}$$

Eq. (2) employs the original $\mathbf{R}$ and $\mathbf{T}$ matrices that describe the baseline geometry, while introducing an additional correction term $\boldsymbol{\Delta}\mathbf{r}_{\text{corr}} \in \mathbb{R}^3$ to obtain the modified global position $\mathbf{r}'_{\text{Global}}$. This can be equivalently expressed, as shown in the Eq. (3), by re-defining the geometry using the updated matrices $\mathbf{R}'$ and $\mathbf{T}'$, in which the correction term $\boldsymbol{\Delta}\mathbf{r}_{\text{corr}}$ is properly absorbed into the updated geometry.

$$\mathbf{r}'_{\text{Global}} = \mathbf{a_R}\mathbf{r}_{\text{Global}} + \mathbf{a_T}. \tag{4}$$

Ultimately, we establish a formalism as shown in Eq. (4) in which the original global position $\mathbf{r}_{\text{Global}}$, computed from the baseline geometry, is transformed into a new global position $\mathbf{r}'_{\text{Global}}$. This transformation incorporates alignment parameters: a rotation matrix $\mathbf{a_R} \in \mathbb{R}^{3\times3}$ and a translation vector $\mathbf{a_T} \in \mathbb{R}^3$.

Our goal is to find $N = 24,120$ sets of correction parameters, $\Theta = \{\mathbf{a_{R,s}}, \mathbf{a_{T,s}}\}_{s=1}^{N_{\text{sensor}}}$, which optimize the physical consistency of the entire system. We denote by $\mathcal{D} = \{\{\mathcal{T}_j\}_{j=1}^{N_{\text{track}}}\}_{i=1}^{N_{\text{event}}}$ the dataset of unlabeled particle trajectories, where each trajectory $\mathcal{T}_j = \{r_{\text{Local}}^{(j,k)}\}_{k=1}^{N_{\text{layer}}}$ is a sequence of local hit positions recorded as the particle traverses a subset of sensors physically installed on distinct detector layers, as shown in Appendix C.

Accordingly, the optimal correction parameters, $\Theta^*$, are obtained by minimizing a physics-based cost function $\mathcal{C}(\Theta)$ evaluated over $\mathcal{D}$.

$$\Theta^* = \arg\min_{\Theta} \mathcal{C}(\Theta). \tag{5}$$

# 4. Methodology: Self-Supervised Learning via Differentiable $\chi^2$ Cost Function

We formulate the alignment problem by constructing 24,120 Neural Networks, each assigned to a specific sensor and optimized to estimate the correction parameters. For a detailed description of the ALICE ITS2 detector geometry and the conceptual mapping of these Neural Network modules, refer to Appendix C and D. Although each network handles only its sensor-wise regression task, the entire module is trained to optimize a single physics-based cost function, ensuring global geometric correctness. This is achieved by utilizing an intrinsic supervision source derived from the physical principle that particle trajectories produced in the same collision, such as pp or Pb–Pb, inherently exhibit smoothness and originate from a common spatial point.

## 4.1. Model Architecture: Single Layer Perceptron

The detector consists of 24,120 individual sensors. Each sensor is assigned a dedicated Neural Network whose role is to extract the correction function for that sensor. The correction function for the s-th sensor is expressed as follows.

$$\Delta \mathbf{r}_{\text{corr},\mathbf{s}} = \mathbf{f}_{\theta_s}(\mathbf{r}_{\text{Local},\mathbf{s}}), \tag{6}$$

where $\mathbf{f}_{\theta_s}(\mathbf{r}_{\text{Local},\mathbf{s}}) : \mathbb{R}^3 \to \mathbb{R}^3$ denotes the Neural Network mapping the 2D local hit position $\mathbf{r}_{\text{Local},s} \in \mathbb{R}^3$ to the 3D global correction vector. Although $\mathbf{r}_{\text{Local},s}$ is a 3D vector, it is sufficient for the Neural Network to take only its two active planar components as input.

Neural Networks are widely recognized as universal approximators capable of representing continuous functions (Cybenko, 1989). Motivated by this fundamental capability, we selected the simplest Neural Network consisting of only one input layer and one output layer, which is sufficient for the linear problem. This is equivalent to a Single Layer Perceptron (SLP), whose architecture in detail is a 2-node input (2D active planar components: $s_1, s_2$) and a 3-node output (3D spatial correction vector: $cs_1, cs_2, cs_3$) with a linear activation as shown in Appendix D. We formulate the correction as a linear relation as follows.

$$\mathbf{f}_{\theta_s}(\mathbf{r}_{\text{Local},\mathbf{s}}) = \mathbf{W}_{\mathbf{s}} \mathbf{r}_{\text{Local},\mathbf{s}} + \mathbf{B}_{\mathbf{s}}. \tag{7}$$

This architecture is not a simplification but an intended design that provides a physics-informed constraint. In principle, the linear mapping of SLPs could also represent translations, rotations, tensile up/down deformations, and shear-like distortions. However, to ensure the model reflects only the 6-DoF physical misalignment of the installed sensors, we impose additional constraints on the weight matrices $\mathbf{W}$ and the bias vectors $\mathbf{B}$.

Thus, the linear formulation in Eq. (7) in our module ensures that the network cannot learn non-physical distortions but is instead constrained to finding the optimal 6-DoF geometric correction for its specific sensor.

To implement this architecture, we developed a custom Neural Network module derived from the TMultiLayerPerceptron class in the ROOT framework (Brun & Rademakers, 1997). This extension allowed us to formulate the cost function and modify the gradient computation mechanisms.

## 4.2. Physics-Informed Self-Supervised Cost Function

The ability of our framework without data labeling is the key Self-Supervised Learning feature. The supervisory signal is provided entirely by a physics-based cost function. We define the cost function as the reduced $\chi^2$ averaged over all $N_{\text{event}}$ collision events, such as pp or Pb–Pb.

$$\mathcal{C}(\Theta) = \frac{1}{N_{\text{event}}} \sum_{i}^{N_{\text{event}}} \frac{\chi^2}{\nu_i}, \tag{8}$$

$$\frac{\chi^2}{\nu_i} = \sum_{j}^{N_{\text{track}}} (\chi^2_{\text{track},j} + \chi^2_{\text{vertex},j}), \tag{9}$$

$$\chi^2_{\text{track}} = \sum_{k}^{N_{\text{layer}}} (\chi^2_{\text{meas,xy}} + \chi^2_{\text{meas,z}})_k, \tag{10}$$

$$(\chi^2_{\text{meas,xy}})_k = \left\{ \frac{1}{\sigma^2_{\text{meas,xy}}} ((r_{\text{xy}} + \Delta r_{\text{xy}}) - \bar{r}_{\text{xy}})^2 \right\}_k, \tag{11}$$

$$(\chi^2_{\text{meas,z}})_k = \left\{ \frac{1}{\sigma^2_{\text{meas,z}}} ((r_{\text{z}} + \Delta r_{\text{z}}) - \bar{r}_{\text{z}})^2 \right\}_k, \tag{12}$$

$$\chi^2_{\text{vertex}} = \chi^2_{\text{vertex,xy}} + \chi^2_{\text{vertex,z}}, \tag{13}$$

$$\chi^2_{\text{vertex,xy}} = \frac{1}{\sigma^2_{\text{vertex,xy}}} (r_{\text{origin,xy}} - \bar{r}_{\text{vertex,xy}})^2, \tag{14}$$

$$\chi^2_{\text{vertex,z}} = \frac{1}{\sigma^2_{\text{vertex,z}}} (r_{\text{origin,z}} - \bar{r}_{\text{vertex,z}})^2, \tag{15}$$

$$\sigma^2 = \sigma^2_{\text{IPR}} + \Sigma_q \sigma^2_{\text{SPP},q}, \tag{16}$$

where $\bar{\mathbf{r}}$ represents the projected trajectory position on the $k$-th sensor plane, $(\mathbf{r} + \Delta\mathbf{r})$ denotes the corrected measurement obtained by applying the learned alignment parameter $\Delta\mathbf{r}(\equiv \Delta\mathbf{r}_{\text{corr},s})$ to the raw sensor hit $\mathbf{r}(\equiv \mathbf{r}_{\text{Local},s})$, $\bar{\mathbf{r}}_{\text{vertex}}$ denotes the point on the fitted trajectory closest to the event origin, and $\mathbf{r}_{\text{origin}}$ corresponds to the common event origin. Each variance ($\sigma^2 \in \{\sigma^2_{\text{meas}}, \sigma^2_{\text{vertex}}\}$) accounts for the total measurement uncertainty, defined as the quadratic sum of the Intrinsic Pixel Resolution (IPR) and the uncertainties induced by Stochastic Physical Processes (SPP). The latter includes multiple scattering and energy-loss fluctuations, which affect the particle trajectory.

This cost function includes two critical components:

1. **Local consistency**: This cost quantifies how well the corrected sensor-level measurements $(\mathbf{r} + \Delta\mathbf{r})$ align with a smooth, fitted trajectory (track). It is the squared sum of residuals, weighted by the measurement uncertainty, $\sigma^2$, which accounts for the physics-based process uncertainty.

2. **Global consistency**: This cost enforces the physical constraint that all trajectories from a single collision event must have originated from a common event origin. This is formulated as minimizing the DCA as shown in Eq. (14) and Eq. (15).

This combined feature creates a self-supervised signal. Any incorrect set of alignment parameters $\Theta$ will cause a high degree of physical inconsistency, manifesting as a high $\chi^2$ cost. The gradient descent algorithm minimizes this inconsistency, guiding the 24,120 networks to find the optimal geometric configuration $\Theta^*$ that satisfies all physical constraints simultaneously.

### 4.3. Optimization

The parameters $\Theta = \{\theta_s\}_{s=1}^{N}$ for all 24,120 networks are updated simultaneously using a gradient-based optimizer. The entire process is end-to-end differentiable. Consequently, the backpropagation algorithm updates the weight matrices $\mathbf{W}$ and bias vectors $\mathbf{B}$ to minimize the cost. Under the implemented constraints, these updates are mathematically equivalent to refining the 6-DoF alignment parameters $\mathbf{a_T}$ and $\mathbf{a_R}$ for each sensor.

The alignment parameters, $\Theta = \{\mathbf{a_{R,s}}, \mathbf{a_{T,s}}\}_{s=1}^{N_{\text{sensor}}}$, are updated by relating the computed gradients from the network parameters, $\Phi = \{\mathbf{W_s}, \mathbf{B_s}\}_{s=1}^{N_{\text{sensor}}}$, through the chain rule. The optimization step at iteration $t$ is defined with learning rate $\eta$ as:

$$\Theta^{(t+1)} = \Theta^{(t)} - \eta \left( \nabla_\Phi \mathcal{C} \cdot \mathbf{J}_{\Phi,\Theta} \right), \quad (17)$$

$$\mathbf{J}_{\Phi,\Theta} = \frac{\partial \Phi}{\partial \Theta}. \quad (18)$$

The Jacobian matrix $\mathbf{J}_{\Phi,\Theta}$ explicitly represents the mathematical constraints linking the network parameters to the 6-DoF physical deformation, ensuring that the minimization is carried out directly in the physical parameter space $\Theta$. While the formulated $\chi^2$ cost function is convex within the region of small misalignments, extending this approach to large deviations would require additional mechanisms to guarantee the global minimum.

### 4.4. Scalable Training via Parallelization

Each sensor is modeled by an individual Neural Network. A single alignment module manages all 24,120 sensor-level networks. Parameter updates are performed trajectory-by-trajectory and strictly sensor-wise, ensuring that each network is updated only by the trajectories that traverse the corresponding sensor.

To scale the training process to massive datasets, we parallelize the alignment module: multiple replicas of the module read separate partitions of the dataset and independently update their sensor-specific parameters. After several epochs, each parallelized module may hold different parameters for the same sensor. This divergence arises because each module processes a distinct data partition. In some cases, a module may not encounter any trajectories for certain sensors, resulting in no parameter updates.

To address this divergence, we periodically synchronize these modules by merging the sensor-wise parameters into a single unified global set as shown in Eq. (19).

$$\theta_s^{(\tau+1)} = \frac{\sum_m \mu_m^{(\tau)} \theta_{s,m}^{(\tau)}}{\sum_m \mu_m^{(\tau)}}. \quad (19)$$

where $s$ denotes the sensor index, $m$ the parallel module index, $\tau$ the synchronization step, and $\mu_m$ the weight of module $m$ determined by the data partition size. This global set is then distributed back to all modules as their new starting state for subsequent parallel updates.

## 5. Experimental Setup

### 5.1. Dataset and Pre-processing

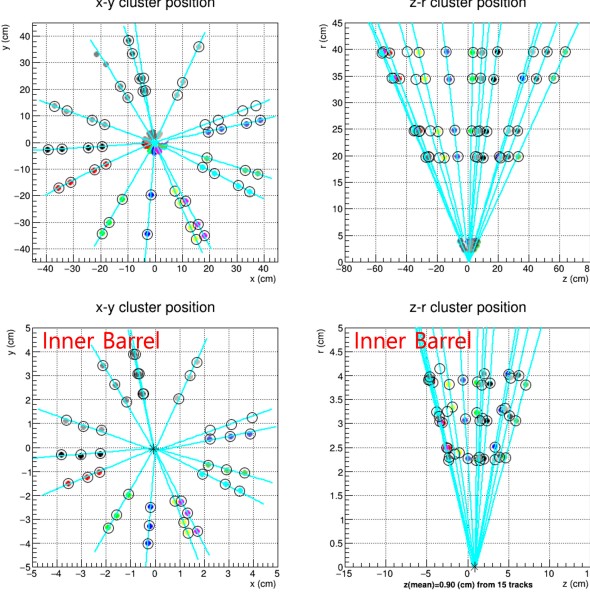

*Figure 1.* Example of reconstructed trajectories from the implemented trajectory finding and fitting system, showing well-separated trajectories.

We use $\sqrt{s} = 13.6$ TeV proton–proton (pp) collision data collected in 2023 by the ALICE experiment at the CERN LHC. The data processing pipeline begins with raw hit (pixel) signals, which are grouped into sensor-level measurements "clusters". These measurements are then processed by a "trajectory finding" and "trajectory fitting" algorithm to identify particle trajectory candidates as shown in Figure 1.

In this study, we only utilize datasets collected under an active magnetic field. Within this field, charged particles follow curved trajectories, enabling the precise determination of their transverse momentum ($p_{\mathrm{T}}$) via the relationship $p_{\mathrm{T}}[\mathrm{GeV}/c] = 0.3B[\mathrm{T}]R[\mathrm{m}]$ (Workman et al., 2022), where $B$ denotes the magnetic field strength, and $R$ the radius of curvature.

For this minimization process, if an event contains multiple trajectories, they are divided into groups of eight trajectories sharing the same collision location, and each group is used as a training input. The training set is used to adjust the network parameters. The validation set is employed for monitoring convergence and stability, while the independent test set is reserved for final performance evaluation.

*Table 1.* Detailed description of data (LHC23zt/539884) that are used in the ML-based alignment process. 1M=1,000,000

| Data set | # of Events | Trajectories per Event |
|---|---|---|
| Train | 5.0 M | 8 |
| Validation | 1.5 M | 8 |
| Test | 1.5 M | 8 |

### 5.2. Implementation Details

The training was performed for approximately two weeks on a CPU-based system. Our framework is implemented in C and C++, built upon the ALICE O2 system (ALICE Collaboration, 2015; ALICE O2 Framework) and the ROOT data analysis framework (Brun & Rademakers, 1997). The full training hyper-parameters, including the optimizer and model architecture, are detailed in the Table 2.

*Table 2.* Training configuration and hyper-parameters of the neural network applied to the module.

| Parameter | Value / Specification |
|---|---|
| NN Architecture | Single Layer Perceptron, (Input: 2, Output: 3) |
| Activation Function | Linear |
| Optimizer | Stochastic Gradient Descent (SGD) |
| Parallelization | 8 Modules ($\mu_m = 1.0$) |
| Framework | ROOT (CERN Analysis Framework), ALICE O2 (Experiment Framework) |
| Hardware (CPU) | Intel i7-14700K |
| Memory Usage | 2GB (Train) / 6GB (Monitor) |
| Training Duration | $\sim$2 Weeks (45 Epochs, $\eta \in 10^{[-12,-9]}$) |

### 5.3. Standard Analytic Baseline

We benchmark our ML framework against the reference alignment geometry. This baseline geometry, $\Theta_{\mathrm{Baseline}}$, was generated using the Millepede algorithm in 2022, a high-performance iterative analytic solver that serves as the standard for detector alignment in high-energy physics.

### 5.4. Evaluation Metrics

We evaluate alignment quality using three complementary metrics: the physics-informed cost $\mathcal{C}(\Theta)$, and the mean and width of DCA distributions. The cost serves as a direct indicator of alignment quality, as it is intrinsically formulated as the residual-based $\chi^2$ quantifying the deviation between reconstructed trajectories and recorded hits from sensors. In parallel, the DCA metric verifies the consistency by measuring the spatial offset between the trajectory and the common event origin.

1. **Physics-Informed Cost**: A residual-based $\chi^2$ is computed over trajectories to quantify system consistency. This explicitly embeds detector-geometry constraints into the learning objective. This depends on the learned parameter set $\Theta^*$ and is evaluated on the training set, validation set, and final test set. During optimization, the training and validation sets are monitored in real time, while the final verification of convergence is performed using the test set. Through this physics-informed cost, the model is steered toward a physically correct and self-supervised alignment solution.

2. **DCA Mean (Bias)**: The mean of the Gaussian-fitted distribution. An ideal alignment achieves a mean of zero.

3. **DCA Width (Resolution)**: The width ($\sigma$) of the Gaussian-fitted distribution. A smaller width signifies a higher-precision alignment. This acts as a composite metric determined by generic noise and alignment quality.

## 6. Results

### 6.1. Improvement of Physics-Informed Cost Function

Our neural network framework successfully optimized the physics-based $\chi^2$ cost. Tables 3 and 4 present the cost components evaluated on independent test sets for the pp and Pb–Pb systems. In particular, the Pb–Pb collision system, characterized by a large multiplicity, i.e. number of produced particles, provides a high-density environment that serves as a robust benchmark for validating geometric consistency under extreme conditions. Therefore, we extended our validation to this dataset to explicitly demonstrate the robustness of our framework.

*Table 3.* Calculated cost with test data (LHC23zt/539884) from pp collisions. (Low trajectory-multiplicity system)

| Geometry | | $\Theta_{\text{Baseline}}$ | | $\Theta_{\text{ML}}$ | |
|---|---|---|---|---|---|
| Multiplicity | Fraction | $\chi^2_{\text{track}}$ | $\chi^2_{\text{vertex}}$ | $\chi^2_{\text{track}}$ | $\chi^2_{\text{vertex}}$ |
| Integrated | 100.0 % | 9.553 | 1.854 | 9.532 | 1.818 |
| 0–9 | 33.9 % | 9.848 | 1.573 | 9.831 | 1.553 |
| 10–19 | 26.7 % | 9.695 | 1.778 | 9.681 | 1.749 |
| 20–49 | 31.9 % | 9.540 | 1.892 | 9.522 | 1.854 |
| 50–99 | 7.4 % | 9.374 | 1.935 | 9.347 | 1.893 |
| 100+ | 0.2 % | 9.212 | 1.961 | 9.148 | 1.918 |

*Table 4.* Calculated cost with test data (LHC23zzo/545345) from Pb–Pb collisions. (High trajectory-multiplicity system)

| Geometry | | $\Theta_{\text{Baseline}}$ | | $\Theta_{\text{ML}}$ | |
|---|---|---|---|---|---|
| Multiplicity | Fraction | $\chi^2_{\text{track}}$ | $\chi^2_{\text{vertex}}$ | $\chi^2_{\text{track}}$ | $\chi^2_{\text{vertex}}$ |
| Integrated | 100.0 % | 9.434 | 2.150 | 9.398 | 2.095 |
| 0–9 | 14.1 % | 9.477 | 1.668 | 9.405 | 1.647 |
| 10–19 | 5.7 % | 9.149 | 1.913 | 9.109 | 1.872 |
| 20–49 | 5.0 % | 9.730 | 2.156 | 9.762 | 2.115 |
| 50–99 | 6.1 % | 9.603 | 2.060 | 9.508 | 2.025 |
| 100+ | 69.1 % | 9.433 | 2.151 | 9.397 | 2.096 |

In both pp and Pb–Pb systems, our method achieved better costs in both local and global consistencies. The most substantial gain was observed in the $\chi^2_{\text{vertex}}$ component, which imposes a common origin constraint on the tracks. Minimizing this component effectively reduces the spatial spread of tracks relative to the estimated primary vertex, directly leading to the observed enhancement in DCA performance.

From a physics perspective, the observed performance metrics are intrinsically linked to the event kinematics of the datasets. The $\chi^2_{\text{track}}$ depends on the track momentum, as the reconstruction uncertainty arises from both the intrinsic sensor precision and stochastic physics processes (e.g., multiple scattering and energy-loss fluctuations) during track propagation. Hence, since the primary vertex is estimated using these tracks, the $\chi^2_{\text{vertex}}$ naturally inherits these kinematic dependencies. Furthermore, the considered track "pulls" the primary vertex, an effect that depends on the multiplicity of particles produced in the event. This mechanism largely explains the small systematic differences in $\chi^2$ values observed between the pp and Pb–Pb systems.

## 6.2. Elimination of Systematic Bias and Enhancement of Pointing Resolution

Improving the DCA performance is a fundamental goal in high-energy physics, as it directly determines the detector's ultimate vertexing precision. We performed an independent validation of the newly determined alignment parameter set, $\Theta_{\text{ML}}$, using the pp collision data collected in 2023.

Figure 2 shows our primary result for the suppression of systematic geometric bias. The proposed ML-optimized alignment (blue) successfully eliminates the mean bias of transverse DCA distribution, resulting in mean values compatible with zero across all $p_{\text{T}}$ (kinematic feature) ranges. In contrast, the baseline alignment (black) exhibits a significant, non-zero bias, especially at low $p_{\text{T}}$.

Figure 3 demonstrates the improvement in resolution. Our ML method (blue) improves the DCA resolution (i.e., achieves a smaller width) compared to the baseline (black) across all $p_{\text{T}}$ ranges. We achieve a resolution improvement of approximately 5–10% in the high-$p_{\text{T}}$ ($> 2\,\text{GeV}/c$) region and 1–3% in the low-$p_{\text{T}}$ region ($< 1\,\text{GeV}/c$).

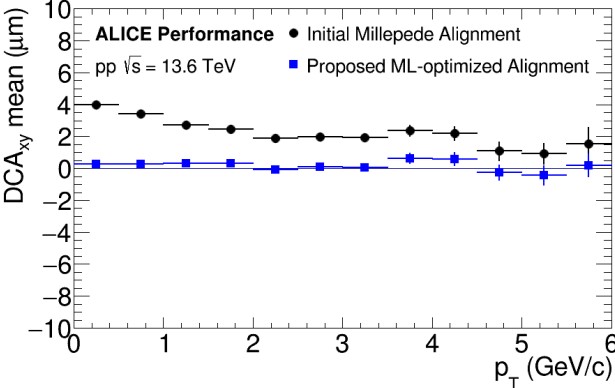

*Figure 2.* Gaussian mean of transverse DCA (DCA$_{\text{xy}}$) as a function of transverse momentum ($p_{\text{T}}$).

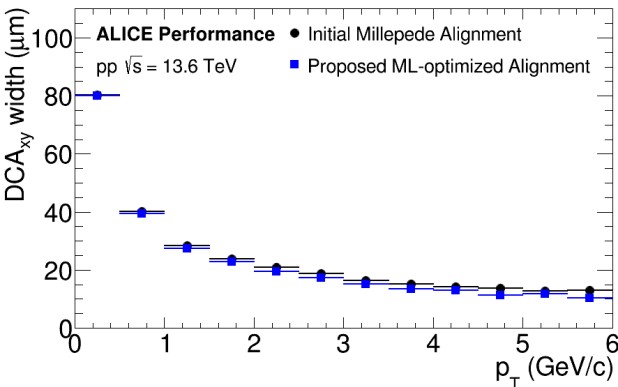

*Figure 3.* Gaussian width of transverse DCA (DCA$_{\text{xy}}$) as a function of transverse momentum ($p_{\text{T}}$).

These two results quantitatively demonstrate that our Self-Supervised ML framework finds a more accurate and unbiased set of geometric alignment parameters $\Theta_{\text{ML}}$ than the previous baseline $\Theta_{\text{Baseline}}$. Consequently, based on these significant improvements, the newly determined alignment set $\Theta_{\text{ML}}$ was successfully applied to the full reconstruction of the ALICE 2023 Pb–Pb dataset.

### 6.3. Analysis: Robustness to Physics-Based Noise

Our analysis identifies why the proposed ML framework demonstrates distinct advantages over the standard method, particularly in the low-$p_\mathrm{T}$ region where data variance is high. The standard approach reduces the statistical weight of low-$p_\mathrm{T}$ trajectories, treating them as unreliable constraints in its linearized least-squares formulation. Low-momentum trajectories experience strong multiple scattering, which induces structured variance of hit residuals that scales inversely with momentum.

In contrast, our ML framework incorporates these low-momentum, high-variance trajectories into the optimization, allowing them to contribute to the fine-tuning of the alignment. This approach demonstrates a robustness to structured physics-based noise. Our end-to-end framework, guided only by the physics-informed $\chi^2$ cost function, correctly models the structured uncertainty arising from heteroscedastic noise caused by multiple scattering. It successfully extracts the faint alignment signal from this high-variance data (the low-momentum trajectories), whereas the standard method largely suppresses such region.

## 7. Discussion

### 7.1. Generalizability to Large-Scale Inverse Problems

Extending beyond the initial application to the ALICE ITS2 detector, our proposed framework addresses large-scale geometric inverse problems characterized by the absence of supervisory signals, while constrained by strict physical consistency. The core potential of our approach, optimizing a scalable array of differentiable affine modules via a physics-informed cost landscape, is applicable to any multi-sensor system satisfying two key conditions: a differentiable geometric parameter space and the presence of intrinsic physical constraints. Potential applications include robotic sensor arrays, multi-camera systems for autonomous driving, and astronomical interferometers.

### 7.2. Future Work: Towards Complex Systems

This work establishes a foundational framework for self-supervised alignment using differentiable lightweight affine transformation modules. Several avenues exist to extend the model's capability and optimization dynamics.

1. **Non-linear Deformation via MLPs**: While the current affine modules effectively identify rigid-body misalignments (6-DoF), as future detector designs evolve towards lightweight or curvilinear architectures, the detector modules may undergo complex non-linear distortions. Upgrading the current affine layers to shallow Multi-Layer Perceptrons (MLPs) would allow the network to approximate these high-order manifolds,

supported by the universal approximation theorem (Cybenko, 1989) to correct flexible deformation.

2. **Inductive Bias via GNNs**: Currently, sensors are optimized independently, neglecting the hierarchical mechanical constraints in detector assemblies. To address this, Graph Neural Networks (GNNs) can be introduced to encode structural dependencies as a relational inductive bias. By representing the detector as a graph, where nodes denote sensors and edges represent mechanical couplings, the networks could effectively learn correlated displacements and propagate geometric corrections across the physical topology.

3. **Adaptive Optimization Landscapes**: This study utilized standard SGD, an effective approach for locally convex landscapes of small misalignments. In contrast, addressing larger-scale and non-linear deformations requires navigating significantly more complex cost landscapes. To effectively correct such substantial misalignments, we propose integrating adaptive optimization algorithms (e.g., Adam (Kingma & Ba, 2015), RMSprop (Tieleman & Hinton, 2012)) as a robust strategy to ensure convergence to the global minimum.

## Impact Statement

We have presented a scalable, self-supervised machine learning framework for large-scale geometric correction. By formulating the system's physical consistency constraints as a differentiable $\chi^2$ cost function, we successfully optimized 24,120 parallel regression Neural Networks using 40 million unlabeled particle trajectories. This approach transforms physics principles into a powerful supervisory signal, enabling our method to determine a more accurate and unbiased solution than the previous standard method. We demonstrated that our framework is highly robust to structured physical noise, effectively extracting alignment constants from high-variance data (low-momentum trajectories) that the standard approach tends to discard.

The successful validation of the proposed alignment geometry by the ALICE experiment confirms the effectiveness of this ML-based approach. This achievement demonstrates the robustness and scalability of our method in a complex, real-world experimental environment, highlighting its potential as a promising next-generation framework for solving high-precision geometric alignment problems in future large-scale systems.

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

## A. Standard Analytic Method: Millepede

The Millepede algorithm serves as the long-established standard for detector alignment in high-energy physics experiments. It formulates the alignment task as a massive linear least-squares minimization problem involving geometrical corrections and particle trajectory parameters.

To solve this high-dimensional optimization, this standard algorithm utilizes the fact that local parameters of different trajectories are uncorrelated, creating a block-diagonal matrix structure in the least-squares normal equations. This algebraic property allows the algorithm to eliminate the massive number of local parameters (Schur complement) and solve a reduced system of equations for the global parameters only. Furthermore, the framework incorporates optimization with linear constraints such as Lagrange multipliers to fix global degrees of freedom or enforce mechanical constraints.

Despite these optimizations, this analytic approach still remains constrained by the computational complexity of the matrix inversion scaling super-linearly with the number of global parameters, resulting in a computationally expensive operation for high-granularity detectors. More critically, the analytic solver requires predefined parameterization of the deformation model based on expert assumptions and manual first-order Taylor expansion. This dependency limits its ability to discover and correct for complex structural distortions that are not predicted by the initial geometric hypothesis.

## B. Distance of Closest Approach (DCA)

The ultimate goal of the ITS2 detector alignment is to improve the vertexing resolution, which is one of the key performance indicators of the ITS2 detector, as well as to minimize the fitting residuals of charged-particle trajectories by correcting the sensor positions.

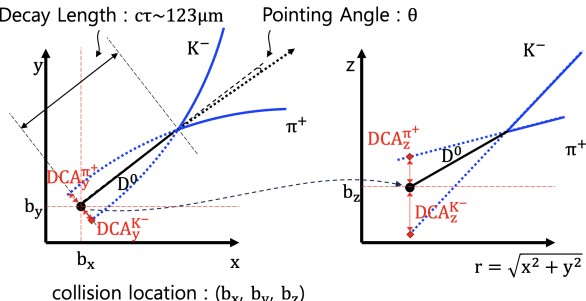

*Figure 4.* Schematic illustration of the $D^0 \to K^- + \pi^+$ decay showing the definition of the DCA in ITS2 vertexing, separated into (left) transverse and (right) longitudinal projections.

Heavy-flavour hadrons containing charm or beauty quarks are characterized by their short lifetimes, resulting in decay lengths typically on the order of hundreds of micrometers. Consequently, the resolution of the Distance of Closest Approach (DCA) is a determining factor for the precise reconstruction and identification of these particles. A representative example is the $D^0$ meson, which contains a charm quark and has a mean proper decay length of $c\tau \approx 123$ μm. In the ALICE experiment, achieving high-precision DCA resolution is essential to distinguish the decay vertices of such short-lived particles from the primary collision vertex.

Using the decay topology of the $D^0$ meson as an example, Figure 4 illustrates the decay process into a kaon and a pion ($D^0 \to K^- + \pi^+$). This topology highlights the importance of precise DCA determination, our evaluation metric, for high-level physics performance, serving as a concrete example of why high-quality alignment is essential. The transverse DCA ($DCA_{xy}$) is defined as the shortest distance between the extrapolated particle trajectory and the primary vertex in the transverse plane. Once the transverse component is determined, the corresponding longitudinal part of the DCA ($DCA_z$) is simultaneously calculated at that point. This calculation method reflects the cylindrical symmetry of the collision geometry in the experiment.

## C. ALICE ITS2 Detector Geometry

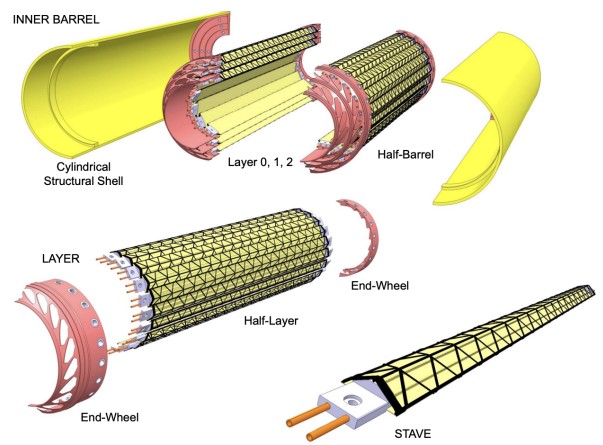

*Figure 5.* Hierarchical geometry of the Inner Barrel (IB)

Figure 5 illustrates the Inner Barrel (IB), which comprises the three concentric innermost layers with the highest pixel granularity. The sensors are arranged on staves, which are then mounted on a cylindrical structural shell, forming the half-barrel geometry.

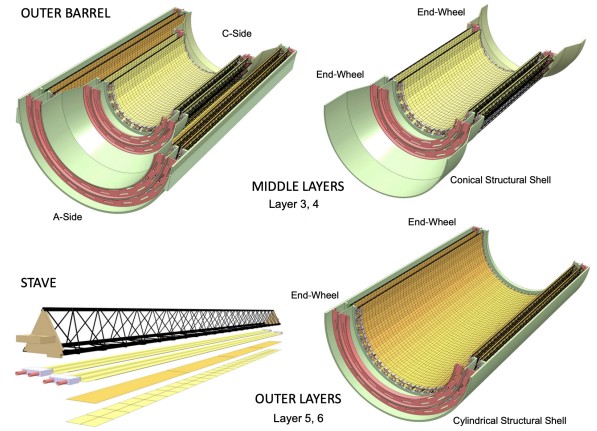

*Figure 6.* Hierarchical geometry of the Outer Barrel (OB)

The Outer Barrel (OB) is divided into the Middle Layers (Layers 3 and 4) and the Outer Layers (Layers 5 and 6). Figure 6 describes the large-scale support structures, including the staves, conical/cylindrical shells, and end-wheels. These hierarchical assemblies, from sensors to staves to layers, impose the mechanical constraints.

## D. Conceptual Design

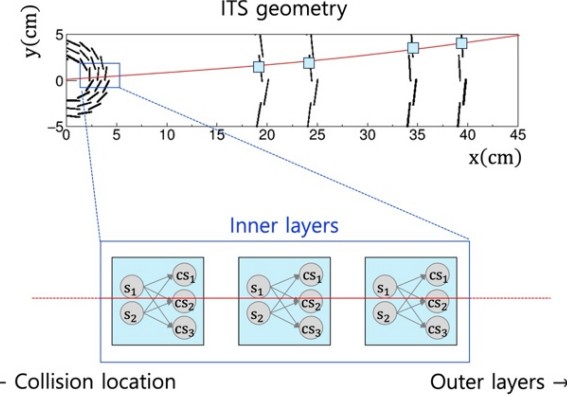

*Figure 7.* Schematic illustration of the ITS geometry and the Neural Network–based ML module design used for detector alignment.

The schematic Figure 7 illustrates a part of the ITS2 detector including three inner and four outer layers, demonstrating how the conceptual NN can be applied to the ITS2 alignment problem. In this approach, an individual NN is assigned to each sensor, allowing an independently trained model to extract its optimal correction parameters.

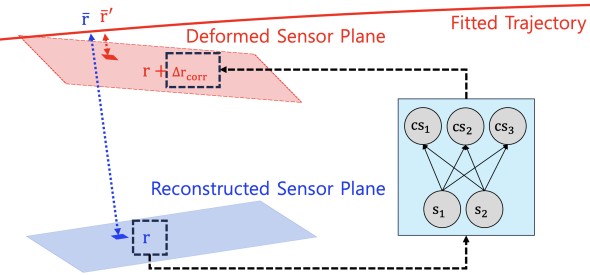

*Figure 8.* Conceptual illustration of the neural alignment framework. The diagram contrasts the ideal reconstructed plane (blue) with the actual deformed plane (red). The sensor-specific Neural Network takes the local hit position $\mathbf{r}$ as input and extracts the correction vector $\Delta\mathbf{r}_{\text{corr}}$, resolving installation discrepancies.

The reconstructed sensor plane represents the 3D spatial mapping of pixel signals derived from the ideal geometry matrix. While this plane would theoretically match the actual particle trajectory under perfect installation conditions, inevitable mechanical deviations necessitate the concept of a deformed sensor plane. Our sensor-specific Neural Networks identify these deformed states to correct subtle discrepancies, thereby improving spatial resolution to meet the Technical Design Report (ALICE Collaboration, 2014) requirements.

