# OpenReview forum: "Self-Supervised Neural Regression for High-Precision Geometric Alignment"
_ICML.cc/2026/Conference — Submitted to ICML 2026_

### Official Review · Reviewer_RsB8 · 2026-03-05

**Soundness:** 3
**Presentation:** 3
**Significance:** 3
**Originality:** 3
**Overall Recommendation:** 4
**Confidence:** 4

**Summary:**

This work introduces a self-supervised learning method for large-scale geometric alignment. The author utilizes the system’s underlying physical constraints as a self-supervised signal, eliminating the need for labeled data. The framework enables parallel optimization of sensor-wise regression models using unlabeled particle trajectories.

**Compliance With Llm Reviewing Policy:**

Affirmed.

**Final Justification:**

I maintain my Weak Accept recommendation: the paper is practically significant and reasonably sound, but its methodological novelty is limited. The rebuttal clarified technical details, but did not change my overall assessment.

**Key Questions For Authors:**

See the questions mentioned above

**Limitations:**

yes

**Strengths And Weaknesses:**

Paper Strengths: The paper is well written, easy to follow, and to understand. A particular strength is the novel self-supervised framework, which derives a robust supervisory signal directly from physics-based consistency requirements, such as trajectory smoothness and vertex compatibility, rather than relying on labeled data. The proposed method leads to a reduced systematic bias and an improved resolution in DCA.
Major Weaknesses:
W1. The core architecture relies on 24,120 single-layer perceptrons (SLPs) with linear activation functions. Beyond its application to large-scale physical systems, what fundamental innovations does this framework offer in network design?
W2. The paper mentions imposing constraints on the weight matrix W and bias vector B of the linear layer (line 111). However, it does not provide details of the constraints. Additionally, how are the orthogonal rotating manifold constraints strictly maintained during the gradient update step?
W3. Performance evaluation only compares the proposed method with the traditional analytical Millepede algorithm. How does it compare to other gradient- or geometry-based registration algorithms (such as ICP or RANSAC variants)?
W4. Training on a CPU-based system takes approximately two weeks (Table 2). While acceptable for offline calibration, this time cost remains high compared to some analysis methods. How much speed improvement is expected if the framework is migrated to a GPU architecture?
Minor Weaknesses:
W1. The discussion and comparison of related work in the related work section is limited. The paper would benefit from a more comprehensive review and contrast with prior methods.
W2. Given that the model assigned to each sensor is essentially a linear regression, is it appropriate to use the term "neural network" throughout the paper?

---

> ### Author Rebuttal · Authors · 2026-03-30
>
> We sincerely thank the reviewer for the thorough and constructive evaluation, and for recognizing the novelty of the self-supervised framework, the practical significance of the results, and the clarity of the writing. We address each concern below.
>
> Major W1.
>
> The linear SLP architecture is a deliberate, physics-informed design choice for the ITS2 detector in ALICE.
> For precision physical alignment, we impose explicit constraints to ensure that the optimization yields only analytically valid 6-DoF solutions, as detailed in Major W2 below.
> Unlike black-box models, every weight $\Phi$ in our network has a direct, physically interpretable mapping with a physical alignment parameter $\Theta$. Our design demonstrates that high-precision scientific tasks benefit from "Glass-box" architectures where the Jacobian provides an explicit, differentiable bridge between neural representation and physical reality.
> More broadly, our work offers the insight that advancing neural network performance does not always require increasing complexity. By leveraging the physical constraints inherent in the problem, a simple architecture can provide physically interpretable solutions. We believe this principle extends beyond our specific application to any domain where well-defined structural constraints exist.
>
> Major W2.
>
> The additional constraints on W and B referenced in Section 4.1 (L111) are enforced by performing the optimization directly in the physical parameter space $\Theta$.
>
> The update procedure is as follows:
> 1. Gradient Computation: Gradients are initially computed in the network parameter space $\Phi_s=\\{W_s,B_s\\}$ using track-cluster residuals in the $\chi^2$ cost.
> 2. Transformation via Jacobian: These gradients are transformed into the 6-DoF alignment parameter space $\Theta_s=\\{\{a_{R,s}, a_{T,s}\}\\}$ using the Jacobian $J_{\Phi,\Theta}$ (Eq. 18).
> 3. Update in $\Theta$-Space: The actual parameter update is performed on $\Theta$, where rotations are represented by three Euler angles ($\alpha, \beta, \gamma$) following the ALICE convention. This ensures that the orthogonality of the rotation matrix is guaranteed by construction at every iteration.
> 4. Mapping back to $\Phi$: The updated $\Theta$ is then mapped back to the network parameters $\Phi$, maintaining a consistent representation of the 6-DoF manifold.
> This mechanism ensures that the optimization only explores physically valid 6-DoF geometric states throughout training. We will explicitly detail this procedure in the revised manuscript.
>
> Major W3.
>
> ICP and RANSAC address structurally different problems from ours.
>
> ICP performs registration of the same object captured from different views by aligning two point clouds through iterative closest-point matching, requiring both an explicit reference geometry and point-to-point correspondences. In our problem, each particle trajectory provides a unique, one-time measurement through multiple sensors.
>
> RANSAC fits the best model from noisy data by randomly sampling point subsets and selecting the hypothesis with the largest consensus. In our problem, all measurements are genuine particle trajectories (hits) with no outliers to reject; the task is to identify geometric deformations from these trajectories.
>
> In the domain of track-based detector alignment, Millepede and its experiment-specific variants constitute the established gold standard across major HEP experiments, making it the most meaningful baseline for evaluating our approach.
>
> We will include a discussion of geometric registration algorithms, including ICP and RANSAC, in the revised Related Work section to clarify the positioning of our approach relative to these methods.
>
> Major W4.
>
> Our framework supports two orthogonal axes of parallelization.
>
> At the CPU level, the training reported in Table 2 used m=8 parallel modules on a single personal computer. We have confirmed stable operation with up to m=400 modules across 400 worker nodes on CERN's GRID infrastructure. Since each module independently processes its partition of the same total dataset, the speedup scales approximately as m_target / m_paper; for m_target=400, this corresponds to a ~50$\\times$ reduction in wall-clock time.
>
> At the data-scanning level, collision events within each module are physically independent, enabling concurrent gradient computation on GPUs. This would provide an additional multiplicative speedup, though the actual factor depends on GPU memory capacity and gradient accumulation overhead.
>
> We will include a detailed discussion of these parallelization experiences in the revised manuscript.
>
> Minor W1.
>
> We will substantially expand the related work section in the revised manuscript to include recent developments in physics-informed machine learning, self-supervised learning, and large-scale distributed optimization, with explicit positioning of our framework relative to these approaches.
>
> Minor W2.
>
> We address this point in detail in our response to Reviewer 7Dpt (Q1).

---

> > ### Author Rebuttal · Reviewer_RsB8 · 2026-04-03
> >
> > I acknowledge the authors' efforts in addressing some of the initial concerns during the rebuttal period. However, my main concerns remain unresolved. The authors argue that the linear single-layer perceptron (SLP) is a deliberate, physics-informed design, and that simplicity itself constitutes an innovation. While I appreciate the value of parsimonious models in applications, this response does not address the core issue: the proposed method lacks substantive algorithmic novelty. Assigning an independent linear regression model to each sensor and optimizing it via gradient descent is mathematically equivalent to solving a large linear least‑squares problem. Therefore, the authors haven't introduced any new representations or optimization strategy in this paper. I would like to keep my recommendation.

---

> > > ### Author Response · Authors · 2026-04-08
> > >
> > > We sincerely thank the reviewer for maintaining the recommendation and for the constructive discussions throughout the review period. We understand and respect the concern on algorithmic novelty.
> > >
> > > The detailed exchanges on the Glass-box design principle, the Jacobian constraint mechanism, and the parallelization strategy have been valuable in sharpening our work.
> > >
> > > We will reflect these discussions in the revised manuscript, with a clearer positioning of this study as a validated milestone in transitioning traditional analytical solvers into a scalable ML framework, one that provides a necessary platform for future HEP experiments demanding even higher-granularity detector alignment.

---

### Official Review · Reviewer_7ba6 · 2026-03-12

**Soundness:** 1
**Presentation:** 1
**Significance:** 1
**Originality:** 2
**Overall Recommendation:** 2
**Confidence:** 3

**Summary:**

The paper proposes a self-supervised, physics-informed learning framework to solve large-scale geometric alignment for a sensor array with per-sensor 6-DoF misalignment, targeting the ALICE ITS2 detector with ~24k sensors (∼150k alignment parameters). The method trains a large number of lightweight sensor-wise regression modules (implemented as constrained linear/affine mappings) by minimizing a differentiable physics-informed 𝜒^2objective that combines (i) local track smoothness/residual consistency and (ii) global vertex-origin consistency via DCA terms.

**Compliance With Llm Reviewing Policy:**

Affirmed.

**Key Questions For Authors:**

- During training, do you re-run track fitting and vertex estimation with updated geometry (and at what cadence), or do you keep trajectories fixed? How does this choice impact convergence and final performance?
- What is the wall-clock time and compute cost of your method vs a standard Millepede workflow for comparable statistics, and how does it scale with number of sensors and tracks?
- How do you handle global degrees of freedom (overall translations/rotations) and mechanical constraints that are typically fixed via constraints/Lagrange multipliers in alignment solvers? Are there modes where the solution is underdetermined or drifts?
- Are the DCA mean/width differences statistically significant (error bars / bootstrapping across runs or subsamples)? What is the variability across different training seeds / partitions?

**Limitations:**

Yes

**Strengths And Weaknesses:**

### Strengths
- Tackles a practically critical, high-dimensional alignment problem at operational scale (24,120 sensors; ~150k parameters) in a real experimental system (ALICE ITS2), which is much more compelling than toy alignment demos.
- The use of a differentiable physics-informed 𝜒^2combining track residual terms and vertex/DCA consistency is a natural and interpretable supervisory signal that avoids manual labels and ties the learning objective to established alignment criteria

### Weakness
- The baseline geometry is described as a Millepede alignment from 2022, while the method is trained/evaluated on 2023 data. Without a Millepede re-alignment on the same 2023 dataset (or a clear justification why that is infeasible), it’s hard to attribute improvements to the proposed method rather than simply to “updated alignment using new data/conditions.”
- The core model is intentionally a single-layer perceptron / affine mapping constrained to represent 6-DoF corrections, trained by minimizing a 𝜒^2. This can be viewed as a gradient-based re-implementation of a classical least-squares alignment objective rather than
introducing a distinctly new ML model class. The paper would benefit from sharper positioning of what is fundamentally new beyond the optimization/engineering pipeline and parallel training scheme.
- The paper claims scalability and reduced CPU time via parallelization, but also reports training taking ~2 weeks on a CPU. A direct runtime/compute comparison vs Millepede (or other established workflows) is important to support the main “scalable alternative” claim
- The paper notes the 𝜒^2landscape is convex only for small misalignments and that larger deviations may need additional mechanisms, but does not characterize how sensitive convergence is to initialization, learning rate schedule (notably extremely small 𝜂), or synchronization frequency.
- Small related work section with few outdated refrences, which give unclear picture about the related work and current challenges, or what is the positioning of this work.

---

> ### Author Rebuttal · Authors · 2026-03-30
>
> We thank the reviewer for the detailed and technically rigorous review. We address the Key Questions first, followed by additional concerns.
>
> Q1.
> Yes. At every epoch, hit positions are updated with the corrected alignment parameters and trajectories are re-fitted accordingly. In principle, the vertex position should also be re-estimated after each track update; however, the resulting change in the re-estimated primary vertex is negligible compared to the per-iteration sensor corrections. We performed one additional vertex re-estimation at epoch 30, where the learning curve (https://anonymous.4open.science/r/icml2026-CCB6/Additional_Figure1.png) had plateaued, and confirmed that the change was also indeed negligible.
>
> Q2.
> A direct wall-clock comparison is challenging because Millepede aligns 192 staves while our framework does 24,120 individual sensors. The ~2-week CPU time corresponds to this sensor-level problem using only a single personal computer.
> Regarding scaling, as discussed in our Introduction and Appendix A, Millepede's computational cost scales super-linearly with the number of parameters due to matrix inversion via Schur complement. In contrast, since each sensor is optimized by an independent module, our framework scales linearly with the number of sensors. Further details are provided in our response to Reviewer RsB8 (Major W4).
>
> Q3.
> In our framework, the $\chi^2_{vertex}$ component (Eqs. 14-15) constrains the global DoF, anchoring the global coordinate frame.
> For rotational DoF, the charge symmetry of produced tracks provides azimuthal stability: any global rotation induces charge-asymmetric DCAxy biases, worsening $\chi^2_{vertex,xy}$. Similarly, tilts about x or y axes induce asymmetric DCAz biases, worsening $\chi^2_{vertex,z}$.
> For translational DoF, DCAs (Eqs. 14-15) directly constrain shifts in x, y, and z. We validated this experimentally (https://anonymous.4open.science/r/icml2026-CCB6/Additional_Figure2.png, blue: paper; green: stability study with X=−30$\mu$m shift). We surveyed various global shifts and tested the most probable candidate considering ITS-TPC track matching residuals. The high-pT DCA bias increased, confirming degraded compatibility with external detectors.
> An additional 2-week stability training showed no drift in sensor-level corrections.
> Regarding mechanical constraints, the residual-based $\chi^2_{track}$ term implicitly enforces consistency. Any non-physical displacement between sensors on the same stave worsens track-cluster residuals.
>
> Q4.
> We validated the statistical significance at two levels.
> First, we performed subsample validation by training on different partitions of the training data and comparing the resulting DCA distributions; the improvements are consistent across these independent runs. This consistency across partitions is precisely what supports the parallelization scheme described in Section 4.4.
> Second, as shown in Tables 3 and 4, the $\chi^2_{vertex}$ improvements are consistent across all multiplicity bins in both pp and Pb–Pb systems, even in two physically distinct collision systems, confirming that the observed improvements are not statistical fluctuations.
> Regarding training seeds: since this work addresses the calibration of a physically installed detector, random initialization would not be meaningful. See also our response to Reviewer 7Dpt (Q2).
>
> Additional concerns
> 1. Baseline fairness: The standard Millepede framework performs alignment at the stave level, 192 staves total (12+16+20+24+30+42+48 across seven layers), with additional parameters for stave deformation functions. Our framework operates at the individual sensor level (24,120 sensors, ~150k parameters), a two-orders-of-magnitude increase in granularity. The improvement cannot be attributed to "updated data" alone. Furthermore, independent attempts to fine-tune using Millepede on 2023 data resulted in divergence of high-pT DCA bias. As illustrated in the figure referenced in Q3 above, while the analytic method (red) achieved internal convergence for low-pT tracks, it suffered from systematic instability in the high-pT region, posing challenges for compatibility with external detectors and limiting its applicability for global tracking. In contrast, our ML framework eliminates systematic biases across the entire momentum spectrum.
>
> 2. Novelty: See our response to Reviewer RsB8 (Major W1), where we discuss the "Glass-box" design principle.
> 3. Computing resources: See our response to Reviewer RsB8 (Major W4).
> 4. Convergence sensitivity: The framework operates in a small-misalignment regime (Eq. 4). The learning rate was experimentally calibrated with an adaptive schedule to prevent divergence. The experimental details including the synchronization frequency are provided in our response to Reviewer 7Dpt (Q2).
> 5. Related work: We will substantially expand the related work section in the revised manuscript. Please see our response to Reviewer RsB8 (Major W3).

---

### Official Review · Reviewer_7Dpt · 2026-03-12

**Soundness:** 2
**Presentation:** 2
**Significance:** 3
**Originality:** 3
**Overall Recommendation:** 3
**Confidence:** 2

**Summary:**

The paper introduces a framework for the high-precision geometric alignment of large-scale sensor arrays, specifically the ALICE ITS2 detector at CERN. The authors replace traditional analytic matrix inversion with a gradient-based optimization approach using 24,120 independent Single Layer Perceptron (SLP) modules. The system is optimized via a differentiable physics-informed $\chi^{2}$ cost function derived from trajectory smoothness and vertex consistency. This approach utilizes physical principles as a self-supervisory signal to determine alignment parameters without manual data labeling.

**Compliance With Llm Reviewing Policy:**

Affirmed.

**Key Questions For Authors:**

- Could the authors clarify the justification for classifying a strictly linear architecture without hidden layers as "Neural Regression" rather than a differentiable affine optimization?
- What specific mechanisms or theoretical insights ensure that the gradient-based optimization consistently reaches a global minimum given the locally convex $\chi^{2}$ cost landscape?

Overall, I value the practical aspects of the paper and will consider raising my rating if my concerns are addressed.

**Limitations:**

yes

**Strengths And Weaknesses:**

**Pros**

* Transitioning from traditional analytic solvers to a gradient-based, parallelizable optimization framework effectively addresses computational and memory bottlenecks in large-scale systems.
* Leveraging intrinsic physical consistency as a supervisory signal enables the model to utilize massive unlabeled datasets while maintaining high spatial precision.
* Successful validation on real-world data from the ALICE experiment demonstrates the practical scalability and robustness of the method in complex experimental environments.

**Cons**

* The authors characterize the framework as "Neural Regression," yet the architecture is strictly restricted to a single layer with linear activations. Mathematically, this model operates as a standard affine transformation, suggesting that the "neural" terminology may significantly overstate the methodological novelty from a deep learning perspective.
* The manuscript lacks a formal proof or rigorous mathematical discussion regarding the convergence of the optimization process. Given the extremely high-dimensional parameter space (150k parameters) and the acknowledged local convexity of the cost landscape, the absence of theoretical grounding for stability is a concern.

---

> ### Author Rebuttal · Authors · 2026-03-30
>
> We thank the reviewer for the thoughtful and constructive review, and for recognizing the practical significance and originality of our work. We address each concern below.
>
> Q1.
> We appreciate this question, as it highlights an important distinction that deserves explicit clarification. The reviewer correctly observes that our per-sensor correction approach is mathematically equivalent to an affine transformation as stated in the Abstract and Section 7. We fully agree. However, this terminology reflects the following methodological aspects of our framework, rather than a claim about architectural complexity.
>
> 1. Intended design
>
> The 6-DoF rigid body correction (3 translations + 3 rotations) that each sensor requires is inherently an affine transformation. We additionally impose explicit 6-DoF constraints on the linear layer to ensure it captures only physically valid rigid-body corrections (e.g. **excluding** tensile and shear-like distortions), as detailed in our response to Reviewer RsB8 (Major W2). Any more complex architecture would further increase the risk of overfitting to non-physical deformation modes.
> - Picture of ALPIDE sensor: https://anonymous.4open.science/r/icml2026-CCB6/Additional_Figure3.jpeg
>
> - Tensile Up: https://anonymous.4open.science/r/icml2026-CCB6/Additional_Figure4.png
>
> - Shear-like distortions: https://anonymous.4open.science/r/icml2026-CCB6/Additional_Figure5.png
>
> The black-outlined white squares represent the nominal sensor positions, while the red squares denote the corrected positions identified by the unconstrained SLP modules. This illustrates the model's ability to capture complex 6-DoF misalignments and various distortions.
>
> 2. Optimization
>
> Unlike traditional analytic solvers that rely on matrix inversion, our framework employs an end-to-end differentiable pipeline where parameters are updated via back-propagation and SGD.
> We also address this point in detail in our response to Reviewer RsB8 (Major W1 and W2).
>
> 3. Future Extensibility
>
> Our framework already supports standard activation functions and configurable hidden layers, enabling a direct upgrade to MLPs for approximating high-order, non-linear deformations in future detector designs.
>
> In summary, the term "Neural Regression" reflects the combination of these three aspects. Our framework was designed not merely to perform affine transformations, but to serve as a general-purpose learning pipeline in which the current linear architecture is one physically motivated configuration, readily extensible to more complex models as future detector designs require.
>
> Q2.
> We provide the following evidence:
>
> 1. Experimental convergence
>
> To ensure the stability of the optimization process, we monitored the cost function (Eq. (8)) on the independent test set (https://anonymous.4open.science/r/icml2026-CCB6/Additional_Figure1.png). These evaluations were performed at every 5-epoch synchronization step ($\tau$). We will include this synchronization parameter in the revised Table 2.
>
> The cost evaluated on the independent test set shows a monotonically decreasing trend, stabilizing at ~1.161 from epoch 25 through the final epoch 45. The alignment parameters at the plateau were treated as candidates and tested through full data reconstruction, including compatibility checks with external detectors. Among these, the epoch 45 version was ultimately selected for final deployment, as the extended training was expected to yield the most refined inter-sensor alignment within the ITS.
>
> 2. Physically Motivated Initial State
>
> In this study, as formulated in Eq. (4) of our manuscript, the framework operates in a small-misalignment regime: the alignment parameters aR, aT represent corrections relative to a pre-existing geometry, not arbitrary initial geometry. Since this work addresses the calibration of a physically installed detector, starting from random initialization would be unnecessary.
>
> 3. Geometric Significance of Radial Paths
>
> In the cylindrical detector geometry (Appendix C and D), inter-sensor coupling arises only between sensors that share particle tracks. Due to the radial propagation from the collision vertex (Figure 1), each sensor is coupled only to other sensors along similar azimuthal and longitudinal directions, not to all 24,120 sensors. This track-defined coupling is inherently sparse, making the optimization landscape significantly more tractable than a fully coupled 150k-dimensional problem.
>
> We hope these clarifications address the reviewer's concerns. We believe the core contribution, demonstrating that physics-informed self-supervision can achieve high-precision detector alignment at operational scale, remains significant.
>
> We will improve the manuscript's clarity on both points in the revised version.

---

> > ### Author Rebuttal · Reviewer_7Dpt · 2026-04-06
> >
> > I appreciate the response, which successfully resolved my concerns for Q2; however, regarding Q1, the linear network design significantly limits the model's architectural novelty—a concern echoed by most reviewers—so I'll maintain my score.

---

> > > ### Author Response · Authors · 2026-04-08
> > >
> > > We thank the reviewer for confirming that the convergence concern (Q2) has been fully resolved. Regarding Q1, we respect the assessment on architectural novelty.
> > >
> > > As noted in our rebuttal, our choice of a constrained, physics-informed architecture was deliberately made to ensure the interpretability and stability required for high-precision physics. As this work is submitted under the Applications area, we believe its primary contribution lies in demonstrating that this framework achieves operationally validated results at a level of granularity that established solvers could not reach. We will ensure this positioning is clearly articulated in the revised manuscript.

---

### Official Review · Reviewer_aF65 · 2026-03-12

**Soundness:** 2
**Presentation:** 1
**Significance:** 2
**Originality:** 3
**Overall Recommendation:** 2
**Confidence:** 3

**Summary:**

A common self-supervised strategy implement on new scenario

**Compliance With Llm Reviewing Policy:**

Affirmed.

**Key Questions For Authors:**

1: the first sentence of 4.3 reads: "The parameters Θ=..for all 24,120 networks are updated simultaneously..." This strongly implies Θ presents the neural network weights/biases.
Then, just two sentences later in the new paragraph, the author writes: "The alignment parameters, Θ=... are updated by relating the computed gradients from the network parameters, " Whether the two Θ  are the same?

**Limitations:**

yes

**Strengths And Weaknesses:**

Strength:
1: A common self-supervised strategy implement on new scenario

Weakness:
1: Almost no key innovation on methodology or algorithm
2: Writing requires improvement
3: Figure visualization is limited

---

> ### Author Rebuttal · Authors · 2026-03-30
>
> **On the $\\Theta$ notation (Key Question)**
>
> We thank the reviewer for flagging this potential point of confusion. The two $\Theta$ are indeed the same quantity.
>
> In Sections 1–3, we introduce $\Theta$ as the correction parameters describing each sensor's geometric displacement.
>
> In Section 4, we establish that these correction parameters are the alignment parameters to be optimized, and Section 4.3 describes how the optimal values are determined.
>
> As shown in Eq. (17), the update is performed directly in $\Theta$-space:
>
> - $\Theta^{(t+1)} = \Theta^{(t)} − η(\nabla_{\Phi} C \\cdot J_{\Phi,\Theta})$
>
> The network parameters $\Phi_s=\\{W_s,B_s\\}$ are updated based on track-cluster residuals in the $\chi^2$ cost. To obtain the 6-DoF alignment parameters $\Theta_s=\\{a_{R,s},a_{T,s}\\}$, the Jacobian $J_{\Phi,\Theta}$ (Eq. 18) is introduced to transform these network parameter updates into alignment parameter updates in the physical parameter space. The actual update occurs on $\Theta$, where rotations are represented by three Euler angles, guaranteeing valid SO(3) rotations by construction. The updated $\Theta$ is then mapped back to $\Phi$ for the next iteration. Please see our response to Reviewer RsB8 (Major W2).
>
> The use of $\theta_s$, rather than $a_{R,s}$ and $a_{T,s}$, in the first paragraph of Section 4.3 was intentionally chosen to maintain notational continuity with Eq. (19) in Section 4.4, where $\theta_{s,m}$ denotes the alignment parameter of the s-th sensor in the m-th parallel module.
>
> We will improve the notation in the revised manuscript.
>
> **On methodology and novelty**
>
> We respectfully believe that the novelty of our work extends beyond applying an existing strategy to a new scenario. We refer to our responses to the other reviewers for a detailed discussion of the framework's contributions, including the physics-informed cost formulation, the sensor-level scalability (24,120 sensors vs. Millepede's 192 staves), and the operational deployment.
>
> We would also like to highlight that our work, submitted to the Applications track, addresses an operationally critical problem where the established analytical solver has reached a structural limitation. The core contribution lies in demonstrating that a physics-informed ML framework overcomes this limitation at production scale, as validated by its official adoption for the LHC Run 3 Pb–Pb data-taking period.
>
> **On writing and visualization**
>
> We acknowledge the concerns on writing quality and figure visualization, and will improve both in the revised manuscript, including a comprehensive schematic illustrating the overall framework.

---

> > ### Author Rebuttal · Reviewer_aF65 · 2026-04-02
> >
> > Thanks for your response.
> >
> > I still have the following questions regarding your work.
> >
> > **Scalability of the Parallelization Strategy**
> >
> > To scale the training process to massive datasets, the authors parallelize the alignment module: multiple replicas of the module read separate partitions of the dataset and independently update their sensor-specific parameters. After several epochs, each parallelized module may hold different parameters for the same sensor. This divergence arises because each module processes a distinct data partition. To address this divergence, the modules are periodically synchronized by merging sensor-wise parameters into a single unified global set. While functional, this synchronization by weighted averaging approach lacks a theoretical convergence guarantee and is not analyzed rigorously. Its relationship to federated learning or data-parallel SGD is not discussed.
> >
> > **Convexity Assumption and Global Minimum Guarantees**
> >
> > While the formulated χ² cost function is convex within the region of small misalignments, extending this approach to large deviations would require additional mechanisms to guarantee the global minimum. This is acknowledged but not addressed, leaving a significant open question about the method's robustness when sensors are far from their nominal positions — a realistic scenario during detector installation.
> >
> > ** Weak Baseline Comparison**
> >
> > The only comparison is against the Millepede-generated geometry from 2022, used as a static reference. There is no ablation demonstrating what happens without the vertex (DCA) term, without the track term, or with alternative ML architectures. The paper would be significantly stronger with:
> >
> > A table showing per-component ablation of the χ² cost function.
> > Comparisons with more recent Millepede runs on the same 2023 data, if available.
> > Inference time benchmarks for both the ML framework and the Millepede baseline.
> >
> > **Training Duration**
> >
> > The training was performed for approximately two weeks on a CPU-based system. This is a very long training time for 45 epochs of gradient descent, and no discussion of GPU acceleration, convergence speed, or training efficiency is provided. For a scalable ML framework, the absence of a runtime analysis is a notable omission.
> >
> > **Generalization Claims Are Premature**
> >
> > The paper claims the framework addresses large-scale geometric inverse problems characterized by the absence of supervisory signals, while constrained by strict physical consistency, and states that the core potential of the approach is applicable to any multi-sensor system satisfying two key conditions: a differentiable geometric parameter space and the presence of intrinsic physical constraints. Potential applications include robotic sensor arrays, multi-camera systems for autonomous driving, and astronomical interferometers. These generalization claims are made without any experimental evidence beyond the single ALICE ITS2 use case.

---

> > > ### Author Response · Authors · 2026-04-08
> > >
> > > We thank the reviewer for the follow-up questions. We address each point below.
> > >
> > > 1.Scalability of the Parallelization Strategy
> > >
> > > Our averaging mechanism (Eq. 19) shares structural similarities with Local SGD / FedAvg, where convergence under IID partitioning and convex objectives is well established. Our setting satisfies the IID condition: collision events are physically independent, and since they are randomly partitioned across modules, each module's gradient for a given sensor is an unbiased sample. However, our system differs in that 24,120 independent per-sensor models are jointly optimized through a shared $\chi^2$ cost with inherent system coupling. We therefore rely on empirical validation: cost decreases monotonically on independent test sets (see our responses to Reviewer RsB8, Major W4 and Reviewer 7Dpt, Q2). A formal convergence analysis incorporating the system coupling will be an important direction for future work.
> > >
> > > [1] McMahan et al., "Communication-Efficient Learning of Deep Networks from Decentralized Data." AISTATS, 2017.
> > >
> > > [2] Stich, "Local SGD Converges Fast and Communicates Little." ICLR, 2019.
> > >
> > > 2.Convexity Assumption and Global Minimum Guarantees
> > >
> > > The framework operates in a small-misalignment regime (Eq. 4): alignment parameters represent corrections relative to a pre-existing geometry from a physically installed detector. Large misalignments are fundamentally outside the scope of any track-based alignment method, including ours. In such cases, track reconstruction itself fails. Our framework addresses high-precision geometric alignment, refining an already functional geometry, not fully automated alignment from an arbitrary initial state. Additionally, the inter-sensor coupling is inherently sparse due to radial track propagation, making the landscape significantly more tractable than a fully coupled 150k-dimensional problem (see our response to Reviewer 7Dpt, Q2).
> > >
> > > 3.Baseline
> > >
> > > Independent attempts to refine using Millepede on the same 2023 data resulted in divergence of the high-pT DCA bias (see our response to Reviewer 7ba6, Additional concern 1, https://anonymous.4open.science/r/icml2026-CCB6/Additional_Figure2.png). Consequently, the 2022 Millepede geometry remains the best-performing analytic baseline available, and our comparison is against this strongest reference, not a weakened one.
> > >
> > > 4.Ablation study
> > >
> > > - $\chi^2$ cost components: the $\chi^2_{track}$ and $\chi^2_{vertex}$ are not alternative design choices but two complementary and independently necessary physical constraints. Removing either term would yield a physically incomplete objective, analogous to fitting a trajectory without requiring it to originate from the collision point, or constraining the origin without ensuring the trajectory is smooth. Accordingly, Tables 3 and 4 in the manuscript report each $\chi^2$ component separately, to verify that both complementary constraints improve simultaneously after alignment correction.
> > >
> > > - Alternative ML architectures: ablation studies serve to disentangle contributions of opaque components in complex architectures. In our Glass-box design, every parameter corresponds to a physically defined 6-DoF rigid-body correction, its contribution to the alignment is fully interpretable by construction, making component-wise ablation redundant.
> > >
> > > 5.Training Duration
> > >
> > > The ~2-week training time corresponds to the full sensor-level problem using only m=8 modules on a single personal computer. This duration reflects the fact that each gradient computation requires a full data scan with per-event track fitting, not a simple forward-backward pass on pre-processed tensors. As detailed in our response to Reviewer RsB8 Major W4, scaling to m=400 on CERN's GRID yields a ~50x wall-clock reduction, and GPU-level parallelization of physically independent collision events would provide an additional multiplicative speedup.
> > >
> > > In the revised manuscript, we will include the comprehensive framework schematic together with a per-step timing breakdown, clarifying the computational cost of each stage (data scan, track fitting, gradient computation, synchronization).
> > >
> > > 6.Generalization
> > >
> > > We agree that the generalization discussion is supported by a single experimental validation. Section 7.1 is explicitly framed as a Discussion of potential applicability, not as an experimentally validated claim. We will revise the language to make this distinction more explicit.
> > >
> > > As a submission under the Applications area, the significance of this work lies not in proposing a fully automated alignment framework, but in demonstrating that ML methodology serves as an effective approach to reach the high-precision vertex resolution required by the Technical Design Report, a performance target that the established analytical solver could not achieve at sensor-level granularity. By adopting the proposed ML approach, we were able to take one step closer to the vertex resolution demanded by the Run 3 physics program (Appendix B).

---

### Decision · Program_Chairs · 2026-04-30

**Decision:**

Reject

**Comment:**

The submission addresses an important applied alignment problem at impressive operational scale, and the reviewers acknowledge its practical relevance, the physics informed objective, and the reported gains on ALICE ITS2. However, the main concerns remain significant: several reviewers view the method as having limited methodological novelty beyond constrained affine optimization, the baseline comparison is not fully convincing, and the paper would benefit from stronger ablations, runtime analysis, and clearer discussion of convergence and related work. The rebuttal helped clarify notation, constraints, and implementation details, but it did not fully resolve the central concerns about novelty and empirical support. Overall, the work is promising from an applications perspective, but it does not yet meet the acceptance bar in its current form.